# Development and Validation of an HPLC-UV Detection Assay for the Determination of Clonidine in Mouse Plasma and Its Application to a Pharmacokinetic Study

**DOI:** 10.3390/molecules25184109

**Published:** 2020-09-08

**Authors:** Haitham AlRabiah, Sabry M. Attia, Nasser S. Al-Shakliah, Gamal A. E. Mostafa

**Affiliations:** 1Department of Pharmaceutical Chemistry, College of Pharmacy, King Saud University, P.O. Box 2457, Riyadh 11451, Saudi Arabia; nalshakliah@ksu.edu.sa; 2Department of Pharmacology and Toxicology, College of Pharmacy, King Saud University, P.O. Box 2457, Riyadh 11451, Saudi Arabia; attiasm@yahoo.com; 3Micro-Analytical Laboratory, Applied Organic Chemistry Department, National Research Center, Dokki, Cairo 12622, Egypt

**Keywords:** clonidine, detection method, HPLC, mouse plasma, pharmacokinetic study

## Abstract

An accurate and simple HPLC-UV method has been developed for the determination of clonidine in mouse plasma. A reversed phase C18 Nova Pack^®^ column (125 mm × 4.6 mm i.d., × 3 μm particle size) was used as stationary phase. The mobile phase composition was a mixture of 0.1% diethylamine/acetonitrile (70:30, *v*/*v*) at pH 8 in an isocratic mode at flow rate was 1.0 mL/min. Detection was set at 210 nm. Tizanidine was used as an internal standard. The clonidine and tizanidine were extracted from plasma matrix using the deproteinization technique. The developed method exhibited a linear calibration range 100.0–2000 ng/mL and the lower limit of detection (LOD) and quantification (LOQ) were 31.0 and 91.9 ng/mL, respectively. The intra-day and inter-day accuracy and precision of the method were within 8.0% and 3.0%, respectively, relative to the nominal concentration. The developed method was validated with respect to linearity, accuracy, precision, and selectivity according to the US Food and drug guideline. Minimal degradation was demonstrated during the determination of clonidine under different stability conditions. The suggested method has been successfully applied during a pharmacokinetic study of clonidine in mouse plasma.

## 1. Introduction 

Clonidine is the imidazoline derivative with the chemical name of 2-[2,6-dichlorophenyl)imino]imidazoline HCl [1], which acts as a hypotensive agent (Figure 1). Clonidine belongs to a class of medications: alpha-adrenergic central agonists [2]. It acts as a hypotensive agent and is used to treat blood pressure disorders, hyperactivity disorders, addiction migraine headaches, and to manage opium and alcohol withdrawal symptoms [3]. Moreover, it decreases the sympathetic nerve activity in the central nervous system [2]. Clinical studies have shown that the changes in clonidine concentration in plasma over time are responsible for its side effects [4]. Therefore, a transdermal patch formulation was developed to maintain the level of clonidine in plasma for long periods of time and to minimize adverse effects [5].

Because of the reported variability in exposure to clonidine, the development of a sensitive and selective procedure for the determination of clonidine level in plasma is required. Various analytical techniques have been reported for clonidine determination by using spectrophotometry [6,7,8], potentiometry [9,10], capillary electrophoresis [11], liquid chromatography-mass spectrometry (LC-MS) [12,13,14,15,16], and gas chromatography-mass spectrometry (GC-MS) [17,18].

Although these hyphenated techniques offer many advantages, such as excellent selectivity and sensitivity, compared with other methods, those techniques are expensive and require highly skilled operators. Therefore, a simple and low cost detection method with a reasonable selectively and sensitivity, will offer wider applicability in various analytical laboratories. Therefore, for the purpose of this study, we chose ultraviolet spectroscopy (UV) as a detection technique with liquid chromatography.

To the best of our knowledge, the use of HPLC-UV detection of clonidine in mouse plasma in a pharmacokinetic study has not been previously reported. The chromatographic run of the proposed method was 8 minutes and the collected were robust, exhibiting a high degree of precision. Finally, applicability of this method in a pharmacokinetic study was demonstrated.

## 2. Results and Discussion

### 2.1. Chromatographic Conditions

HPLC-UV detection is widely used in quality control, pharmaceutical analysis, and clinical application [19,20,21], which provided acceptable sensitivity and low coast in comparison with more advanced techniques (e.g. HPLC-MS or GC-MS). Therefore, HPLC-UV detection was used for the assay of clonidine in mouse plasma. Tizanidine was used as an internal standard (Figure 1). Clonidine and tizanidine were detected at 210 nm. The chromatographic conditions were optimized (selection of stationary phase and mobile) to get good separation conditions. The reversed phase C18 Nova Pack^®^ column (125 mm × 4.6 mm (internal diameter) × 3 μm particle diameter) as an analytical column (Water) was used for separation of the investigated drug. The most suitable mobile phase was a mixture of 0.1% diethylamine/acetonitrile (70:30, *v*/*v*) adjusted to pH 8 (using phosphoric acid) in isocratic mode. The chromatographic separation was carried out at 25 °C at a mobile flow rate of 1.0 mL/min. The run time was 8 min and the elution times of IS and clonidine were 5 and 6.5 min, respectively. Table 1 show the system suitability parameters of the developed HPLC method.

### 2.2. Extraction Recovery

A sample clean-up procedure must be applied to the plasma before the HPLC investigation. The sensitivity and selectivity of the proposed strategy can be improved with the clean-up step. To avoid the loss of analyte during the clean-up step and to enhance the clean-up, extraction steps should be kept to a minimum, and the most suitable method would be a one-step procedure. Protein precipitation is regularly utilized as a quick clean-up method and a way to overcome protein-medication binding. In the present study, acetonitrile was chosen as the protein precipitant because it gave a high recovery of clonidine and removed most of the protein species present in the plasma. The purpose of the deproteinization strategy in the proposed test methodology is the elimination of the need for multi-step extraction techniques, such as those required in solid-phase extraction and fluid extraction.

The protein precipitation procedure using acetonitrile extraction of clonidine from mouse plasma is presented in Table 2. The extraction recovery was in the range of 93.5–105% with relative standard deviation (RSD%) in the range 0.29–2.24%. The results indicate that the protein precipitation procedure shows a good efficiency for the extraction of clonidine from mouse plasma.

### 2.3. Specificity of the Developed Method

Six different plasma samples were monitored to record the selectivity of the proposed method. The results for drug-free mouse plasma, which used as blank and mouse plasma spiked with 200 and 1000 ng/mL of clonidine and IS, respectively are presented in Figure 2. The peaks of IS and clonidine were well eluted at a retention time of 5.0 and 6.5 min. No peaks from the drug-free plasma were found to overlap with peaks of either clonidine or tizanidine. The total run time was less than 8.0 min for clonidine determination; the sample deproteinization process was suitable for extracting clonidine and IS from plasma without extracting any interfering species with similar retention time.

### 2.4. Validation of the Method

#### 2.4.1. Linearity and Sensitivity

In accordance with food drug administration (FDA) guidelines [21] the suggested procedure was validated with regard to accuracy, precision, calibration range, limit of quantification (LOQ), and limit of detection (LOD) before application of the proposed method for routine subject analysis. The chromatographic parameters containing retention time, selectivity, peak asymmetry, and resolution were tested to investigate the efficiency of the chromatographic method. The suitability parameters results are summarized in Table 2.

A good relationship was demonstrated between the peak area ratios for clonidine to IS and the related clonidine concentrations (ng/mL) over the concentration range of 100–2000 ng/mL. The linear regression equation of the peak area ratio (*y*) versus the drug concentration in the mouse plasma samples (*x*) showed a good correlation coefficient (r^2^) = 0.999 (*y* = 0.002x − 0.0944). The value of the correlation coefficient and the standard deviation of the slope and intercept of the calibration curve are expressions of the good linearity of the calibration curve [22,23].

Table 3 show the analytical characterization parameters of the calibration graph. The LOQ and LOD were 91.9 and 31.0 ng/mL, respectively, at signal-to-noise ratios of 10 (LOQ) and 3.3 (LOD), respectively [23]. The accuracy of the proposed method was in the range of 93.5–108.0%, while the precision was less than 2%.

#### 2.4.2. Accuracy and Precision

The accuracy and precision of the proposed HPLC method in mouse plasma in the concentration range of 100–2000 ng/mL was examined [22,23,24]. Four quality control samples were analyzed for each concentration ( number of repation *n* = 3) during a single day and on different days. The intra-day and inter-day accuracy and precision for clonidine were calculated from the results. The accuracies calculated from the calibration graph were in the range of 96.3–111.5% and 93.3–109.22% for the intra-day and inter-day results, respectively. The precision was in the range of 0.25–1.7% and 0.36–1.96% for the intra-day and inter-day, respectively. The results are presented in Table 4.

On the other hand, the accuracy of the quality control samples were in the ranges of 92.1–96.0% and 90.0–104.0% for the intra-day and inter-day results, respectively, whereas the precision was recorded in the range of 0.1–2.3% and 0.21–2.43% for the same sequence, respectively. The results are presented in Table 5.

#### 2.4.3. Stability

The QC samples of clonidine were used to study the stability of clonidine under different conditions (Benchtop to 8 h, stability at −4 °C and thaw stability). The benchtop stability for the QC samples for 8 h showed recoveries of 107.5%, 96.3% and 97.1%, respectively, whereas the RSD% values were 1.55%, 1.02%, and 0.78% respectively. The stability at −4 °C was evaluated for the QC concentrations for 24 h; the results showed recoveries of 110.2%, 96.5%, and 97.5% respectively, whereas the RSD% values were 1.53%, 1.22%, and 0.7% respectively.

In addition, the thaw stability of the QC samples was evaluated for three cycles each. The recovery values were 104.5%, 95.0%, and 95.5%, whereas the RSD% values were 1.34%, 0.86%, and 0.88%. The results are presented in Table 6. These studies indicate that there was a high degree of accuracy and precision in the determination of clonidine under different stability conditions.

#### 2.4.4. Ruggedness

The ruggedness of the investigated method was assayed with different instruments and analysts. The results obtained with two different operators and two different instruments were found to be respectable. Relative standard deviation values were less than 2.5% for repetitive measurements. The results indicated that the suggested procedure is highly rugged.

### 2.5. Application to PK Studies

The investigated procedure was approved for the determination of clonidine in mouse plasma. The concentration of clonidine in mouse plasma were determined at an interval time after administration. A concentration–time curve (AUC) of clonidine was drawn, as shown in Figure 3. The pharmacokinetic parameters of clonidine were assessed from the concentration–time graph after the administration of 2 mg/kg clonidine. The mean estimation of T_max_ and C_max_ were 2 h and 135 ± 14.11 ng/mL, respectively. No other chromatographic peaks appeared at the elution time of clonidine or IS (Figure 4). This study was constructed using the HPLC-UV method, and this equipment is more readily available in most laboratories compared with HPLC-MS or GC-MS instruments.

## 3. Experimental

### 3.1. Reagents and Materials

Clonidine and tizanidine (IS) (>99% purity) as the reference standard were obtained from Sigma-Aldrich (St. Louis, MI, USA). The structures of clonidine and tizanidine are illustrated in Figure 1. Acetonitrile, methanol (HPLC-grade), and diethylamine of analytical reagent grade were obtained from BDH (Poole, UK). Ultra-pure water was obtained from a Milli-Q plus purification system (Millipore, Waters Milford, MA, USA).

### 3.2. Apparatus

HPLC analysis was carried out on a Waters HPLC system (Milford, MA, USA) equipped with a 1500 series HPLC pump. The Empower pro chromatography manager was used for data acquisition analysis. The separations were completed with an analytical C18 Nova Pack^®^ column (125 mm × 4.6 mm internal diameter × 3 μm particle diameter) manufactured by Waters, USA coupled with an asymmetry C18-sentry guard column (20 mm). The mobile phase was filtered through a 0.45 μm Millipore system and degassed with ultrasonication. A dual-wavelength ultraviolet detector (2489) and an autosampler (717plus) were used.

### 3.3. Chromatographic Conditions

The mobile phase was a mixture of 0.1% diethylamine and acetonitrile (70:30, *volume*/*volume*, *v*/*v*) pH 8.0. The mobile phase was prepared daily, filtered, and degassed. The separation process was performed in isocratic mode at a flow rate of 1 mL/min at 25 °C. The detection wavelength was set at 210 nm and the injection volume was 50 μL.

### 3.4. Preparation of Solutions

Stock solutions of clonidine and tizanidine (IS) (1 mg/mL) were prepared by an appropriate dilution of an exact amount of each compound in methanol. Two working solutions of drug and IS were prepared by transferring 2.5 mL of stock solution into 25 mL of methanol to obtained a 100 µg/mL solution (WI) and then diluting this again to obtained a 10 µg/mL solution (WII). An additional working solution of 1 µg/mL of clonidine was prepared by transferring 2.5 mL from the second working solution into 25 mL of methanol in a measuring flask (WIII). All solutions were kept in a refrigerator and were stable without any degradation for at least for two months.

### 3.5. Preparation of Clonidine Calibration Curve

Six concentrations of clonidine (100, 200, 400, 800, 1000, and 2000 ng/mL) were prepared in 2.0 mL of disposable microcentrifuge tubes by spiking of 100 µL of mouse plasma with clonidine and internal standard. A 500 µL aliquot of acetonitrile was added to each tube as the deproteinization solvent, after which each tube was vortexed for 1 min and then centrifuged for about 12 min at 10,000 rpm. A supernatant liquid was filtered through a 0.45 μL Millipore filter; then, 50 microliters of the clear solution were injected into the HPLC system. Blank mouse plasma samples were treated with methanol instead of clonidine and IS; then, they were treated as a sample (vortexed, centrifuged, filtrated through Millipore filter), and clean solution was injected into the HPLC system.

Four quality control samples of different concentrations of 150, 550, 1250, and 1750 ng/mL were prepared. A direct calibration graph was created by plotting the peak area ratio (clonidine to IS) (*y*-axis) against the concentration of clonidine (*x*-axis). A linearity equation was extracted from the constructed calibration graph. The identical parameters of the calibration: e.g., slope, intercept, and coefficient of variation (r^2^) were calculated.

### 3.6. Validation of the HPLC-UV

Validation of the proposed HPLC-UV detection was examined according the US Food Drug and Administration guidelines [21,24]. The different criteria of validation parameters were assessed.

#### 3.6.1. Recovery

The recovery is defined as how much of the initial concentration from the calibration curve was recovered. The recovery of clonidine in spiked mouse plasma was tested at concentrations of 100, 200, 400, 800, 1000, and 2000 ng/mL by comparing the concentration determined for the six extracted samples with the concentration of clonidine added.

Four quality control (QC) samples at different concentrations of 150, 550, 1250, and 1750 ng/mL were also prepared and assessed by the proposed method. To evaluate any interference in the investigated method, drug-free plasma was examined as a blank (Equation (1)).
(1)$$Recovery,\%=\frac{Foundconcentration}{Addedconcentration}\times 100$$

#### 3.6.2. Precision and Accuracy

Intra-day and inter-day accuracy and precision of the proposed HPLC-UV procedure were investigated by testing different concentrations of clonidine in the calibration range, during the same day and on different days [21]. Accuracy is defined as the closeness of a measured value to the true value and is established by comparing the peak area ratios of extracted samples with peak area ratios of reference samples; accuracy is expressed as percent recovery. Precision is the repeatability of the measurement based on a standard solution on the same day and over the different days; precision is expressed as percent RSD (Equation (2)).
(2)$$RSD=\frac{SD}{Mean}\times 100$$

#### 3.6.3. Stability

The stability of clonidine was examined in mouse plasma by analysis of a quality control sample on the same day (kept on the benchtop for 8 h). The stability of QC samples kept at −4 °C for 24 h was evaluated after three freeze/thaw cycles with storage at −80 °C and thawing to 25 °C. The mean, recovery, standard deviation, and RSD% values were estimated from the pre-constructed calibration curve.

### 3.7. Pharmacokineitc Study

#### 3.7.1. Animal Testing

A pharmacokinetic (PK) study was conducted with 10–12-week-old Swiss albino mice (weighing approximately 25–30 g) obtained from the Experimental Animal Care Center, Faculty of Pharmacy, King Saud University. Mice were kept in our laboratory for 2 days under standard conditions, standard humidity, temperature, illumination, diet and water (12 h light–dark cycle). All animal experiments were conducted in accordance with the legal requirements in “Saudi Arabia for experiments with animals and humans”. The institutional review board of our university, as previously published [24], approved this study with the ethical approval No: SE-19-102.

#### 3.7.2. Clonidine Pharmacokinetic in Mice

After 7 days of settlement, the mice were arbitrarily partitioned into seven groups comprising of 3 mice each. Six gatherings of mice were orally treated with 2 mg/kg of clonidine, and the remaining groups of mice were treated with regulated saline to give the clear drug-free mouse plasma. The administrated volume was 0.01 mL per g body weight. Clonidine was administrated 0.5, 1, 2, 3, 4, 5, and 10 h before blood testing. In each case, blood samples were drawn from the mouse, and plasma samples were isolated from the blood by centrifugation and then kept at −20 °C until investigation. The maximum plasma concentration of clonidine (C_max_) and the time to reach that point (T_max_) were calculated from the plasma concentration–time curve.

## 4. Conclusions

An accurate and precise HPLC-UV method for the determination of clonidine in mouse plasma was established. Chromatographic separation was carried out on a reversed-phase C18 column. A mixture of 0.1% diethylamine and acetonitrile (70:30, *v*/*v*); pH 8 was used as the mobile phase at a flow rate of 1 mL/min in isocreatic mode. Tizanidine was used as the internal standard. The sample was extracted by using acetonitrile as a deproteinization solvent. The sugussted method was validate according to FDA guidelines. The current HPLC method has been used for the assay of clonidine in mouse plasma. No degradation was observed during the determination of clonidine with this method; the short- and long-term stability of the drug were unaffected. The proposed method demonstrated high precision, accuracy, and sensitivity for use in pharmacokinetic studies of clonidine.

## Figures and Tables

**Figure 1 molecules-25-04109-f001:**
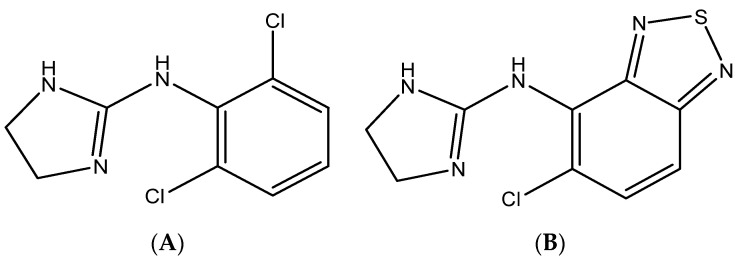
Chemical structure of (**A**) clonidine and (**B**) tizanidine (IS).

**Figure 2 molecules-25-04109-f002:**
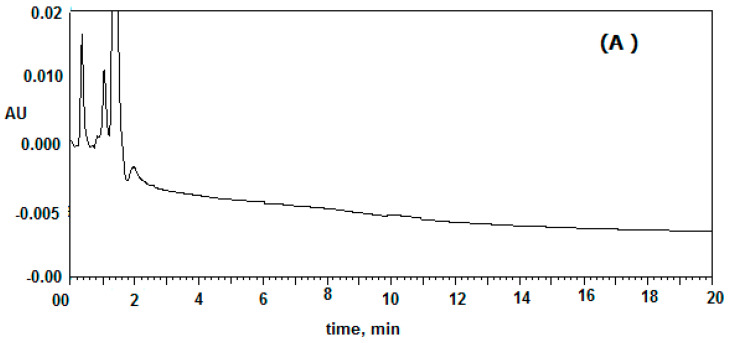
HPLC chromatogram of clonidine: (**A**) mouse plasma, (**B**) mouse plasma spiked with 1 μg/mL (Tizanidine, IS) and 200 (ng/mL) of IS and clonidine respectively (retention time was 5.002 and 6.505 (min)).

**Figure 3 molecules-25-04109-f003:**
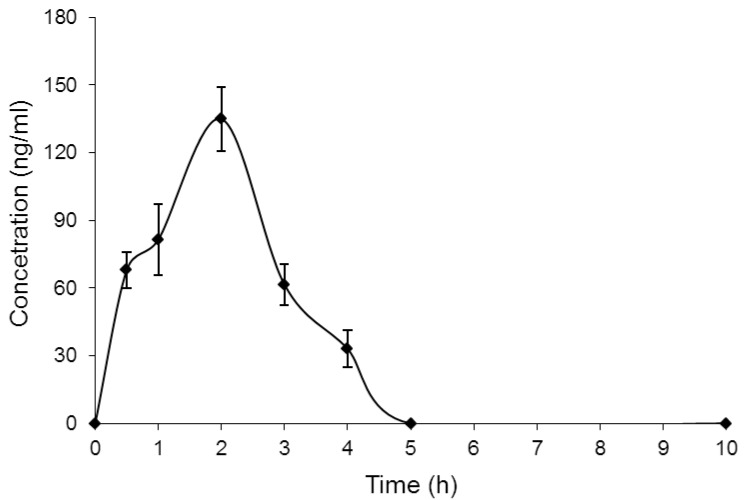
The concentration–time curve of clonidine in mice plasma after administration with 2 mg/kg of clonidine (each point equal mean ± SD).

**Figure 4 molecules-25-04109-f004:**
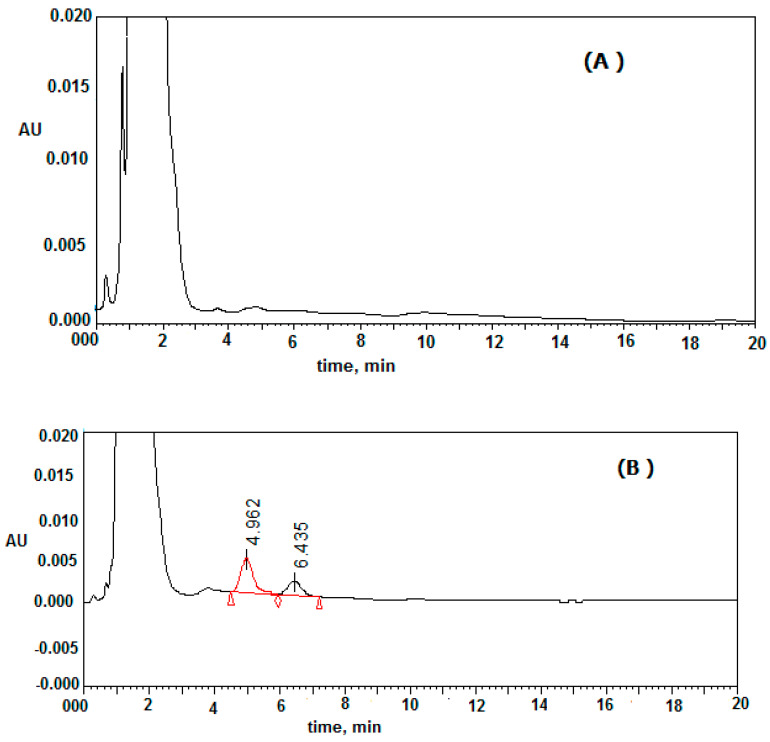
HPLC chromatograms of mouse plasma sample after 3 h from administration (**B**) and (**A**) bank mouse plasma.

**Table 1 molecules-25-04109-t001:** System suitability parameters.

Parameters	RT *	K	Selectivity	Resolution	Symmetry Factor
IS	5.0	89.48	1.5	-	1.06
Clonidine	6.5	131.61	2.47	2.91	1.0

* RT = retention time.

**Table 2 molecules-25-04109-t002:** Extraction of clonidine from mice plasma using the proposed HPLC method.

Spiked (ng/mL)	Found ^a^ (ng/mL)	Recovery (%) ^b^	RSD *, %
100.0	105.0	105.0	0.29
200.0	216.0	108.0	0.69
400.0	380.0	95.0	2.24
800.0	748.0	93.5	0.87
1000.0	960.0	96.0	0.52
2000.0	2060.0	103.0	0.32

^a^ Clonidine extracted from mice plasma; ^b^ Average recovery (repeatation = 3) (*n* = 3). * RSD, relative standard deviation.

**Table 3 molecules-25-04109-t003:** Analytical characteristic parameters of the investigated HPLC method. LOD: limit of detection, LOQ: limit of quantification.

Parameter	Clonidine ^a^
Slope	0.002
The standard error of the slope	2.85 × 10^−5^
Intercept	−0.0944
The standard error of intercept	0.03142
S_Y/X_ *	0.0168253
Correlation coefficient, (r^2^)	0.999
Calibration range (ng/mL)	100.0–2000.0
LOD (ng/mL)	31.0
LOQ (ng/mL)	91.9
Retention time for clonidine (min)	6.5
Retention time for IS (min)	5.0

^a^ Values are the mean of three determinations.* S_Y/X_ is Standard error of Y/X.

**Table 4 molecules-25-04109-t004:** Accuracy and precision of clonidine in spiked mice plasma using the proposed HPLC method.

Conc. (ng/mL)	Intra-Day				Inter-Day			
	Mean	SD	RSD%	Accuracy %	Mean	SD	RSD%	Accuracy %
100	109.0	0.25	0.25	109.10	107.0	0.45	0.42	107.0
200	223.0	1.31	0.58	111. 5	211.0	1.55	0.73	105.5
500	481.6	8.5	1.7	96.3	478.0	9.4	1.96	95.6
1000	969.0	6.7	0.69	96.9	960.0	7.5	0.78	96.0
1500	1445.0	4.95	0.34	96.3	1400.0	5.1	0.36	93.3
2000	2068.5	6.4	0.31	103.4	2184.3	8.1	0.37	109.22

(number of repation *n* = 3).

**Table 5 molecules-25-04109-t005:** Accuracy and precision of the quality control samples by the proposed HPLC.

Conc. (ng/mL)	Intra-Day				Inter-Day			
	Mean	SD	RSD%	Accuracy %	Mean	SD	RSD%	Accuracy %
150.0	138.1	3.2	2.31	92.1	136.0	3.3	2.43	90.6
550.0	520.0	2.5	0.11	94.5	575.0	2.4	0.42	104.54
1250.0	1187.5	1.37	0.16	95.0	1168.75	2.4	0.21	93.5
1750.0	1680	1.38	0.10	96.0	1464.5	1.9	0.13	94.0

(number of repation (*n* = 3).

**Table 6 molecules-25-04109-t006:** Stability data of clonidine in mouse plasma.

Parameters	Bench Top for 8 h			Stability at −4 °C			Thaw Stability		
	Conc. ng/mL			Conc. ng/mL			Conc. ng/mL		
	150	550	1250	150	550	1250	150	550	1250
Mean	161.3	529.7	1213.8	165.3	530.7	1206.3	156.8	522.5	1193.8
Recovery *, %	107.5	96.3	97.1	110.2	96.5	96.7	104.5	95	95.5
SD	2.5	5.4	9.5	2.53	6.5	8.5	2.11	4.5	10.52
RSD, %	1.55	1.02	0.78	1.53	1.22	0.7	1.34	0.86	0.88

* Average of three determinations.

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
