# Peer review of "Development and Validation of an HPLC-UV Detection Assay for the Determination of Clonidine in Mouse Plasma and Its Application to a Pharmacokinetic Study"

_molecules, 2020, doi:10.3390/molecules25184109_

Round 1

Reviewer 1 Report

The article Development and Validation of an HPLC-UV Detection Assay for the Determination of Clonidine in Mouse Plasma and its Application to a Pharmacokinetic Study is well written, it involves relevant literature, experimental part is detailed, results are well discussed. I recommend it for publication after following changes:

Page 2/line70: “Tizanidine of concentration 1000 ng/ml was used as internal …”

Table 1: do not use shortcuts in Table 1, delete “Added con.” use: “Spiked concentration” or only “Spiked”, since there are units, it is clear, that it is concentration, similarly as in the case “Found” one think of concentration.

Page 3/line 102: Specify “a therapeutic concentration of clonidine”, the value or range of concentrations. The sentence is not clear.

Fig. 3: in the text, where Fig. 3 is described, it should be specified, how these data were measured. All values except one are below limit of quantification LOQ. Possibly, spiking with lower concentrations 30 and 50 ng/ml was required?

Page 10/line 250: Do not write in the heading “PK”, write “Pharmacokinetic study”

Page 10/line 252: write “The pharmacokinetic (PK) study…”

Author Response

Report 1

Q1

Page 2/line70: “Tizanidine of concentration 1000 ng/ml was used as internal …”

Answer

Tizanidine 1000 ng/ml is correct as internal standard and its concentration was modified as 1mg/ml

Q2

Table 1: do not use shortcuts in Table 1, delete “Added con.” use: “Spiked concentration” or only “Spiked”, since there are units, it is clear, that it is concentration, similarly as in the case “Found” one think of concentration

Answer

Shortcuts in table 1 was deleted only spiking and Found are used

Q3

Page 3/line 102: Specify “a therapeutic concentration of clonidine”, the value or range of concentrations. The sentence is not clear

Answer

The sentence has been modified to ………….The total run time was less than 8.0 min for clonidine determination.

Q4

Fig. 3: in the text, where Fig. 3 is described, it should be specified, how these data were measured. All values except one are below limit of quantification LOQ. Possibly, spiking with lower concentrations 30 and 50 ng/ml was required.

Answer

The concentrations 30 and 50 ng/ml  in Fig. 3  are still in the LOD range:  30 ng/ml  is the lower limit  of detection which can be detected with increasing the repeating of the experiment ( n=7), whereas 50 ng/ml was upper the LOD , which can be detected very easy

Q5

Page 10/line 250: Do not write in the heading “PK”, write “Pharmacokinetic study”

Answer

Pharmacokinetic study was inserted instead of PK

Q6

Page 10/line 252: write “The pharmacokinetic (PK) study…”

Answer

The pharmacokinetic (PK) study… has been inserted instead of PK”

Reviewer 2 Report

The manuscript “molecules-918470” presents a procedure to analyze clonidine in mouse plasma using an HPLC method with UV detection.  The manuscript has value describing a simple but reliable method for the measurement of the analyte.  Several improvements would be necessary before publication:

-          The  manner the manuscript is organized should follow a different scheme: 1. Introduction, 2. Experimental, 3. Results and Discussion, 4. Conclusions.  In the Experimental part, should be included besides those present in the manuscript, also parts included by the authors in “Results and Discussion”.  The “Experimental” should include subsection 2.1, 2.2 where sample preparation such as the deproteinization procedure should be detailed, the specificity (2.3), and the whole validation part (these are mentioned in the manuscript in two places).

-          The authors indicate that the mobile phase used for separation was adjusted to pH 8 (L 75).  How was the adjustment made? 

-          The Results and Discussion should describe the Application to PK study and whatever other comments the authors have.

-          The Results and Discussion must include comparison with other procedure for the analysis of clonidine.

Author Response

Report 2

Q1

The  manner the manuscript is organized should follow a different scheme: 1. Introduction, 2. Experimental, 3. Results and Discussion, 4. Conclusions.  In the Experimental part, should be included besides those present in the manuscript, also parts included by the authors in “Results and Discussion”.  The “Experimental” should include subsection 2.1, 2.2 where sample preparation such as the deproteinization procedure should be detailed, the specificity (2.3), and the whole validation part (these are mentioned in the manuscript in two places).

Answer

The manuscript has been organized according the following scheme: 1.introduction, 2.Results and discussion, 3.experimental and 4. Conclusion part.  Also the experimental part have been subsection (2.1, 2.2, 2.3, 2.4, and so one ) and validation part have been inserted into two section (experimental part and results and discussion part).

Two references have been changed its number 19, 20 converted 23, 24  and references 21, 22, 23 and 24  have been re-numbered 19, 20, 21, and 22

Q2

The authors indicate that the mobile phase used for separation was adjusted to pH 8 (L 75).  How was the adjustment made? 

Answer

The adjustment of the mobile phase was adjusted to pH 8 by orthophosphoric acid

Q3

The Results and Discussion should describe the Application to PK study and whatever other comments the authors have.

Answer

The section 2.5 has been inserted (PK study) in the results and discussion part , in additional section  3.7.1 and 3.7.2  ( PK study) have been inserted.  

Q4

The Results and Discussion must include comparison with other procedure for the analysis of clonidine.

Answer

In the results and discussion part, a comparison with the other procedure for the analysis of clonidine has been inserted.
